# Antimicrobial Use in Animals in Timor-Leste Based on Veterinary Antimicrobial Imports between 2016 and 2019

**DOI:** 10.3390/antibiotics10040426

**Published:** 2021-04-12

**Authors:** Shawn Ting, Abrao Pereira, Amalia de Jesus Alves, Salvador Fernandes, Cristina da Costa Soares, Felix Joanico Soares, Onofre da Costa Henrique, Steven Davis, Jennifer Yan, Joshua R. Francis, Tamsin S. Barnes, Joanita Bendita da Costa Jong

**Affiliations:** 1Global and Tropical Health Division, Menzies School of Health Research, Charles Darwin University, Ellengowan Drive, Darwin, NT 0909, Australia; abrao.pereira@menzies.edu.au (A.P.); amalia.dejesusalves@menzies.edu.au (A.d.J.A.); steven.davis@menzies.edu.au (S.D.); jennifer.yan@menzies.edu.au (J.Y.); josh.francis@menzies.edu.au (J.R.F.); 2Ministry of Agriculture and Fisheries, Government of Timor-Leste, Av. Nicolao Lobato, Comoro, Dili 0332, Timor-Leste; baduhlareanatena@gmail.com (S.F.); cristinasoares@gmail.com (C.d.C.S.); romakabureno@gmail.com (F.J.S.); henriqueonofre11@gmail.com (O.d.C.H.); katitadog_2001@yahoo.com (J.B.d.C.J.); 3Department of Paediatrics, Royal Darwin Hospital, Darwin, NT 0810, Australia; 4Epivet Pty. Ltd., Withcott, QLD 4352, Australia; tamsin.barnes@gmail.com

**Keywords:** antimicrobial use, antimicrobial resistance (AMR), Timor-Leste, antibiotic, antimicrobial, veterinary, prudent use, critically important antimicrobials, growth promotion, poultry

## Abstract

Monitoring veterinary antimicrobial use is part of the global strategy to tackle antimicrobial resistance. The purpose of this study was to quantify veterinary antimicrobials imported into Timor-Leste between 2016 and 2019 and describe the antimicrobial import profile of importers. Data were obtained from import applications received by the Ministry of Agriculture and Fisheries (MAF) of Timor-Leste. Import quantities were analysed by antimicrobial class, importance for human medicine, recommended route of administration and type of importer. An average of 57.4 kg (s.d. 31.0 kg) and 0.55 mg/kg (s.d. 0.27 mg/kg) animal biomass of antimicrobials was imported per year. Tetracyclines (35.5%), penicillins (23.7%), and macrolides (15.9%) were the commonly imported antimicrobial classes. Antimicrobials imported for parenteral administration were most common (60.1%). MAF was the largest importer (52.4%). Most of the critically important antimicrobials for human medicine were imported by poultry farms for oral administration and use for growth promotion could not be ruled out. In conclusion, the use of antimicrobials in animals in Timor-Leste is very low, in keeping with its predominantly subsistence agriculture system. Farmer education, development of treatment guidelines, and strengthening of the veterinary service is important for addressing the potential future misuse of antimicrobials especially in the commercial poultry industry.

## 1. Introduction

The emergence of antimicrobial resistance is a major global health threat for the 21st century [1]. It is also a One Health challenge that requires coordinated action as transmission of resistant bacteria can occur between humans, animals, plants and the environment [2,3,4]. This emergence has been rapid and is linked to the overuse and misuse of antimicrobials in humans and animals [5,6]. Despite this, it is projected that the use of antimicrobials in humans and animals will continue to rise over the next decade [7,8]. In particular, the use of antimicrobials in food producing animals has received attention due to high levels of use globally for disease prevention and growth promotion [9,10]. While some developed countries have demonstrated a reduction in usage levels [11,12,13,14,15], usage in many developing countries have risen due to farm intensification and demand for animal-based protein associated with rising incomes [16,17,18]. This puts low- and middle-income countries at a higher risk for emergence of resistance.

Antimicrobial resistance limits the effectiveness of antimicrobial therapy which has a greater impact in low and middle-income countries due to their weaker health systems, higher prevalence of infectious diseases and limited access to more expensive treatment alternatives [19,20]. To preserve the effectiveness of antimicrobials, a global strategy has been developed to tackle antimicrobial resistance [21]. This strategy is wide-ranging and multi-sectoral and includes initiatives to strengthen monitoring of antimicrobial use in animals [21].

Monitoring of antimicrobial use in animals at the national level enables a country to identify trends of use over time and assess the impact of policy measures to promote prudent use in animals [22]. When analysed in conjunction with data on antimicrobial resistance in animal and humans, it can also identify potential associations between antimicrobial use and resistance patterns [23,24]. To harmonize antimicrobial use data collection, the World Organisation for Animal Health (OIE) has published guidelines for monitoring the use of antimicrobials in food producing animals [25]. The guidelines acknowledge that antimicrobial use data can be obtained from different levels such as import, manufacturing, sales, dispensing records or from end-use sources [25]. While many higher income countries have been collecting data for many years [26,27], some low to middle income countries in Africa and Asia-Pacific are still facing challenges such as a lack of regulation, under-reporting and unreliable data when monitoring antimicrobial use in animals [10,28,29].

Timor-Leste is a lower-middle income country [30] located in the south-east portion of the Malay Archipelago with a population of 1.3 million [31]. Subsistence farming is the main livelihood for most of the rural population [32,33], with a high proportion of households owning livestock [34]. Chicken and pigs are the two most commonly reared species in the country [35]. Commercial animal farming is uncommon [36,37] but may increase with rising income levels [33]. Currently, there are two large commercial layer farms [38] and a growing number of commercial broiler farms. There are no major commercial livestock farms for other species. There is no local manufacture of antimicrobials, and all antimicrobials are imported into the country. All applications to import veterinary medicines into the country must be submitted to the Ministry of Agriculture and Fisheries (MAF) and there is no re-export of veterinary antimicrobials.

The aims of this study were to quantify veterinary antimicrobial imports into Timor-Leste between 2016 to 2019; and to describe these imports based on antimicrobial class, importance for human medicine, recommended route of administration and type of importer. The findings can help improve monitoring and control of veterinary antimicrobial use in Timor-Leste.

## 2. Materials and Methods

### 2.1. Data Collection for Antimicrobial Imports

All applications to import veterinary medicines into Timor-Leste submitted to MAF between January 2016 to December 2019 were screened to identify veterinary antimicrobials using OIE’s list of antimicrobials of veterinary importance [39]. Data on the date of application, name of importer, brand name, quantity imported, name of active ingredient, concentration of active ingredient, route of administration and target species was extracted for each veterinary antimicrobial. Any missing details on the name of active ingredient, concentration of active ingredient and route of administration was obtained from the technical product sheets. Data collection was performed by two MAF staff who received training on recording antimicrobial import data from received import applications through three workshops and ongoing side-by-side mentorship [40]. The data was stored on an Excel spreadsheet (Microsoft Corporation, Redmond, WA, USA). Data accuracy was checked independently by three researchers from Menzies School of Health Research between November and December 2020.

### 2.2. Data Categorisation for Antimicrobial Imports

Using the name of the active ingredient, each antimicrobial was classified into an antimicrobial class/subclass based on OIE guidelines [41]. The name of the active ingredient was also used to classify antimicrobials as a critically important antimicrobial (CIA), highly important antimicrobial (HIA) or an important antimicrobial (IA) using the World Health Organization (WHO) List of Critically Important Antimicrobials for Human Medicine [42]. The importer name was used to classify importers into 6 types to understand their individual import patterns: “MAF”, “agriculture shops”, “veterinary clinics”, “layer farms”, “broiler farms”, and “education institutions”. Layer and broiler farms were placed in separate categories because they may have different antimicrobial use patterns. In Timor-Leste, agriculture shops are enterprises where veterinary medicines can be procured without a prescription.

### 2.3. Animal Biomass Calculation

Data for biomass calculation (i.e., number of live animals, number of animals slaughtered and meat product quantity) were obtained from the Food and Agricultural Organization Global Statistical Database (FAOSTAT) [43,44]. Common animal species in Timor-Leste (buffalo, cattle, chicken, goats, horse, pigs, and sheep) [34] were included in the biomass calculation. Ducks, rabbits, dogs, and cats were excluded because data were not available. Total animal biomass was calculated for each year between 2016 and 2019 using an OIE method [25] except for bovine biomass because the proportion of animals in different age groups are not known. The data for total animal biomass calculation and estimates for annual biomass can be found in Appendix A.

### 2.4. Data Analysis

The weight of active ingredient in one unit of imported product per pharmaceutical form (e.g., bottle, bag, or tube) was estimated by multiplying the strength of the antimicrobial active ingredient by the volume or weight. All weights were expressed in kilograms (kg). Conversion factors based on OIE guidelines was used to mathematically convert international units (IU) into kilograms [29].

The weight of each active ingredient imported between 2016 to 2019 was calculated by multiplying the weight of active ingredient in one unit of product by the quantity imported. Adjustment for animal biomass was achieved by dividing the total weight of active ingredient by the total animal biomass. The result was expressed in milligram (mg) of active ingredient per kilogram (kg) of animal biomass.

Annual and total imports were calculated for each active ingredient, antimicrobial class, WHO class of importance in human medicine, route of administration and type of importer. Total annual imports of all antimicrobials by weight and weight adjusted for biomass were summarized as mean ± s.d. Spearman’s rank correlation coefficient (r_s_) was used to test the hypothesis of a monotonic (increasing or decreasing) trend in imports by total weight, total weight adjusted for biomass, individual active ingredient and type of importer. Data analysis was performed using Stata 15 software (StataCorp, College Station, TX, USA)

### 2.5. Ethical Approval

The study was conducted in accordance with the Declaration of Helsinki, and the protocol was approved by Human Research Ethics Committee of the Northern Territory (NT) Department of Health and Menzies School of Health Research (2020-3841) and Institute Nacional de Saude in Timor-Leste (MS-INS/DE/IX/2020/1411).

## 3. Results

### 3.1. Import Quantities and Trends

Between 2016 to 2019, a total of 229.8 kg of active ingredients of veterinary antimicrobials were imported into Timor-Leste (mean: 57.4 ± 31.0 kg per year). Import quantities were lower in 2017 and 2018 compared to 2016 and 2019 (see Table 1). After adjusting for animal biomass, the average amount of imported antimicrobials was 0.55 ± 0.27 mg/kg biomass per year. There was no evidence of a significant monotonic trend in antimicrobial imports based on total weight (r_s_: −0.40, *p* value: 0.60) or weight adjusted by biomass (r_s_: −0.40, *p* value: 0.60) (see Figure 1).

A total of 21 antimicrobial active ingredients belonging to 8 classes of antimicrobials were imported during the study period. The import quantities of different antimicrobials between 2016 and 2019 can be found in Table 1. The active ingredients imported in the largest quantities were oxytetracyline (81.7 kg; 35.5%), amoxicillin (34.8 kg; 15.2%), tylosin (25.2 kg; 11.0%) and dihydrostreptomycin (25.8 kg; 11.2%). The classes of antimicrobials imported in the largest quantities were tetracyclines (81.7 kg; 35.5%), penicillins (54.4 kg; 23.7%), macrolides (36.5 kg; 15.9%) and aminoglycosides (25.8 kg; 11.3%). There was some evidence of monotonic increase in imports of neomycin (r_s_: 0.95, *p* value: 0.05) but quantities imported each year were extremely small. There was also some evidence of a monotonic decrease in imports of tylosin (r_s_: −0.95, *p* value: 0.05) driven by a relatively large import in 2016 and sulfamonomethoxine (r_s_: −0.95, *p* value: 0.05) although quantities imported each year were extremely small. There was no strong evidence of a monotonic trend in the import of any of the other individual antimicrobials (see Table 1). Based on WHO classification, most of the imported veterinary antimicrobials were CIAs (117.9 kg; 51.3%) followed by HIAs (111.8 kg; 48.7%).

### 3.2. Import Pattern by Recommended Route of Administration

Recommended routes of administration for imported veterinary antimicrobials during the study period were parenteral (138.0 kg; 60.1%), oral (91.5 kg; 39.8%), and topical (0.3 kg; 0.1%). The majority of tetracyclines (81.3 kg; 99.6%), aminoglycosides (25.8 kg; 99.9%), sulphonamides (13.4 kg; 96.2%), and cephalosporins (0.01 kg; 100%) were for parenteral administration, while the majority of penicillins (37.6 kg; 69.2%), macrolides (36.3 kg; 99.5%), polypeptides (10.9 kg; 100%), and fluoroquinolones (6.0 kg; 100%) were for oral administration. The quantities of different antimicrobial classes for parenteral, oral and topical administration are shown in Figure 2. The weight of antimicrobial classes recommended for administration through different routes for each year over the study period can be found in Appendix A.

### 3.3. Import Pattern by Importer Type

Between 2016 and 2019, the biggest importers of antimicrobials were MAF (120.4 kg; 52.4%), followed by layers farms (81.1 kg; 35.3%) and agriculture shops (15.9 kg; 6.9%) (See Figure 3). There was very strong evidence of a monotonic increase in antimicrobial imports by MAF (r_s_: 1.0, *p* value: <0.001) and evidence of a monotonic increase in antimicrobial imports by broiler farms (r_s_: 0.95, *p* value: 0.05) but no evidence of a monotonic trend in antimicrobial import patterns for other types of importers (see Figure 4A,B). The pattern of imports by layer farms was unique as imports were high in 2016 (58.6 kg) and 2019 (22.5 kg) but negligible between those years (see Figure 4A). Educational institutions imported a relatively small amount (0.6 kg) of antimicrobials once in 2016. Colistin, neomycin and enrofloxacin were only imported by layer or broiler farms. Cephalosporins were only imported by veterinary clinics. The weights of individual antimicrobials and antimicrobial classes imported by different type of importers for each year during the study period can be found in Appendix A.

The biggest importers of CIAs were layer farms (81.1 kg), MAF (25.6 kg) and broiler farms (9.0 kg). Layer and broiler farms imported CIAs almost exclusively; while CIAs accounted for less than a quarter of imports by MAF, agriculture shops and veterinary clinics (see Figure 5). Almost all antimicrobial imports by layer and broiler farms were for oral administration; while almost all imports by MAF and agriculture shops were for parenteral administration (see Figure 6).

## 4. Discussion

### 4.1. Strengths of the Study

This is the first study to describe veterinary antimicrobial imports into Timor-Leste. It showed a very low level of antimicrobial use in animals. Future studies of a similar nature will enable analysis of long-term trends and identification of changes in import patterns arising from interventions. Import data is a reasonable proxy for actual antimicrobial use for Timor-Leste since there is no local manufacture of veterinary antimicrobials and no re-export of antimicrobials. The data collection method was implemented consistently as it was performed by trained personnel using a written protocol. The accuracy of data was checked rigorously by authors to minimise data entry errors, and calculations were done with methods aligned with international guidelines. The training during data collection strengthened the capacity of MAF personnel to record antimicrobial import data and facilitated the timely reporting of results to the OIE, which is often a challenge in developing countries.

### 4.2. Quantity of Antimicrobial Import

The quantity of antimicrobials imported for use in animals in Timor-Leste after adjusting for biomass (0.55 mg/kg biomass) is very low compared to the global average of 144.39 mg/kg and regional average (Asia, Far East, and Oceania) of 237.72 mg/kg in 2016 [29]. The use of veterinary antimicrobials in Timor-Leste was even lower than countries such as New Zealand, Norway, and Iceland which are known to have some of the lowest use levels in the world [45,46]. The low level of use is likely due to the subsistence agriculture system in Timor-Leste [33,47] where there is poor access to veterinary services and medicines. The low level of use is also consistent with another study in Timor-Leste which showed that only 1% of backyard chicken farmers used commercial medicines in their animals [48]. It would be interesting to compare the results from Timor-Leste to other countries with a similar agriculture background but similar studies from such countries could not be found [10]. Although antimicrobial use levels are currently low, use may increase in the future with farming intensification, as seen in other developing countries [49,50]. In this study, there is already evidence of increasing use in the broiler industry, with import levels rising by 119% between 2018 and 2019.

### 4.3. Trend of Antimicrobial Import

Trends in antimicrobial imports over the study period can be explained by looking individually at each importer. For MAF, the rise in antimicrobial imports during the study period represented increased procurement following annual feedback that government employed animal health professionals (e.g., veterinary and livestock technicians) faced shortages for field use [51].

For layer farms, it is likely that the import quantities were inconsistent between years because this group included only two large commercial layer farms that import antimicrobials in bulk quantities for use over a few years. For broiler farms, antimicrobial imports occurred only after 2018 following the import of day-old chicks from Indonesia after the lifting of avian-influenza related import restrictions [52]. The easing of restrictions was followed by a government effort to promote the growth of the broiler industry. The use of antimicrobials may also reflect the lack of resources to implement farm biosecurity and vaccination programmes on these farms [53,54]. Use of antimicrobials on broiler farms could be expected to rise in the future, mimicking the trends seen in neighbouring Indonesia where there was a rise of antimicrobial use due to industry growth, lack of alternative disease control options and a relatively low cost of antibiotics [55]. Therefore, farmer education programmes to improve knowledge on good animal husbandry practices and biosecurity could be useful [56]. The availability of quality vaccines would provide further options for disease prevention and control [57].

For agriculture shops, the reason for a decrease in imports during the study period was unclear but could be partially attributed to non-adherence to the MAF import application process resulting in data not being captured. For veterinary clinics, the low quantities imported reflect the small size of the industry—there were only four veterinary practices operating in Timor-Leste during the study period. The closure of one veterinary clinic in 2018 coincided with a drop in antimicrobial imports by veterinary clinics that year. For education institutions, there was only a once off import of antimicrobials by an agriculture school in 2016. There were no direct imports of antimicrobials by other types of commercial livestock farms apart from poultry, but animals on these farms could still receive antimicrobials imported by MAF or agriculture shops.

### 4.4. Antimicrobial Class and Importance for Human Medicine

The common antimicrobial classes in Timor-Leste (tetracycline, penicillin, and macrolide) are consistent with global and regional (Asia, Far East, and Oceania) usage patterns [29]. The most imported antimicrobials in Timor-Leste (oxytetracycline, amoxicillin, tylosin, and dihydrostreptomycin) were consistent with antimicrobials used in poultry and pig production in developing countries in Asia and Africa [18,50,58,59]. Oxytetracycline is popular because of its broad-spectrum action, low cost, and availability in long-acting formulations [60,61] and it is likely that similar reasons underpin its popularity in Timor-Leste. Amoxicillin and tylosin were imported almost exclusively in oral formulation by commercial poultry farms, and the popularity of these antimicrobials in small scale poultry farms were also reported in other studies in other countries [50,62]. Dihydrostreptomycin was commonly imported in formulations with benzylpenicillin by MAF due to the combination’s broad-spectrum action across a wide range of livestock species. It was positive that colistin, which is an antibiotic of last resort for human medicine that is commonly used in developing countries [63,64] contributed to less than 5% of imports to Timor-Leste with the majority imported in 2016. However, the broiler industry has been importing colistin albeit in small quantities in recent years and this should be closely monitored.

There has been a strong push towards reducing the use of medically important antimicrobials in livestock globally [65]. The almost exclusive imports of CIAs by commercial poultry farms could be attributed to the lack of awareness on antimicrobial resistance and its impact on public health, which has been observed in studies elsewhere [66,67]. On the other hand, the low proportion of CIA imports by MAF (21.2%) and veterinary clinics (21.7%) puts the professional veterinary service in positive light in terms of preserving critically important antimicrobials for use in human health. Of important concern is the import of fluroquinolones, polymyxins, and 3rd and 4th generation cephalosporins which are highest priority critically important antimicrobials for human medicine. Although the combined quantity of these classes contributed to less than 8% of total imports, future import and distribution of these antimicrobials should be closely monitored because of the potential risk of the development and transmission of antimicrobial resistance from livestock to humans [68,69]. To address the high proportion of CIA usage in the commercial poultry sector, a jointly developed antimicrobial treatment guideline between government and industry preferencing the use of non-CIA antibiotics may be effective [70].

### 4.5. Route of Administration

In this study, antimicrobials recommended for oral administration (39.8%) were less common than reported in some countries [12,14,71]. Antimicrobials imported by MAF and agriculture shops were mostly for parenteral administration. This is likely to be because they were mainly for use in species such as pigs and cattle that are reared extensively on small-holder livestock farms. On the other hand, commercial poultry farms probably imported mainly antimicrobials for oral administration because they are convenient for mass administration in poultry reared in semi-intensive or intensive production environments. The use of orally administered antimicrobials should be monitored in Timor-Leste as it has been demonstrated elsewhere that this route is more prone to misuse from inappropriate dosing and promotes the development of antimicrobial resistance [72].

### 4.6. Use of Antimicrobials for Growth Promotion

The import of antimicrobials intended for oral administration raises the concern of use of antimicrobials for growth promotion. Ideally antimicrobials should not be used for growth promotion without a public health risk assessment and any use should be phased out especially for critically important antimicrobials [21,65]. According to MAF, antimicrobials are not known to be used for growth promotion in the country. However, oral bacitracin and tylosin that were imported by poultry farms have been used for growth promotion worldwide [28,73]. In addition, the technical fact sheet of some antimicrobials indicated that the products could be administered for growth promotion. Therefore, it is possible that commercial farmers are administering antimicrobials at low doses, as recommended for growth promotion, without being aware. This has also been reported in another study [49]. The possible use of antimicrobials for growth promotion in Timor-Leste should be further investigated.

### 4.7. Use of Antimicrobials in Aquaculture

Although antimicrobial use is common practice in aquaculture systems worldwide and regionally [74,75,76], there were no aquaculture importers in this study and no products were indicated for use in aquatic animals. The absence of antimicrobial use in this sector is likely due to the relatively small and underdeveloped aquaculture sector [77].

### 4.8. Limitations

Timor-Leste is not immune to the challenges and limitations of monitoring antimicrobial use. The study only included import data after 2016 because of the five-year holding limit of hardcopy applications in the MAF office and the lack of digital record keeping. Thus, only data from 2016 to 2020 were available. The study excluded data from 2020 because the calendar year of 2020 had not yet ended at the point of data collection. This short study period limited the power of the study to detect trends in import quantities. However, digital record keeping was initiated as part of the study which will enable future studies to cover a longer time period.

It is likely that the study provided an under-estimation of the total amount of antimicrobials used in Timor-Leste due to the non-submission of import applications by some importers as reported in another study [71]. The possible reasons for non-submission include an importer’s desire to avoid waiting times for approval, a weak regulatory framework and the lack of enforcement. Antimicrobials intended for human use could have also been administered to animals although elsewhere this is usually limited to companion animals [78]. The authors predict that the underestimation would not result in more than a doubling in the total amount of imported antimicrobials during the study period. Even if this happened, Timor-Leste would still demonstrate one of the lowest use rates compared to other countries that have reported usage data.

There may be a small degree of inaccuracy for the animal biomass estimation because the data obtained from FAOSTAT was based on extrapolations. This source of data was used because annual census data was unavailable during the study period. The OIE method for biomass calculation involved the use of European conversion coefficients and breeding cycles that may be different to Timor-Leste. However, these default parameters were used as no suitable alternative for a Timor-Leste context was found. The exclusion of minor species such as ducks, rabbits, dogs, and cats from the biomass calculation is likely to have only a marginal impact on the result since the population is relatively small [25].

It was not possible to quantify the antimicrobials that were administered to different animal species based on the import data due to the multi-species indication for many of the antimicrobials. However, a rough estimation of the division of antimicrobial use between livestock and companion animals could be estimated by assuming that antimicrobials imported by veterinary clinics were administered exclusively to companion animals, and antimicrobials imported by all other importers were administered exclusively to livestock.

### 4.9. Future Directions

Although the use of antimicrobials in animals is Timor-Leste is very low, there is potential for future misuse and overuse with farming intensification. Future studies investigating the knowledge, attitudes and practices of animal health professionals and farmers on antimicrobial use would be useful for identify strategies for promoting prudent use of antimicrobials in animals as identified in other studies [79,80,81]. Even in the absence of such studies, early action can be informed by studies conducted in other developing countries [55,58,82]. In addition to farmer education, which was mentioned previously, improving farmers access to animal health professionals [64], and training of animal health professionals to engage with farmers on prudent antimicrobial use has been shown to be effective elsewhere [82]. The strengthening of laboratory capacity in bacterial culture and antimicrobial susceptibility testing will also facilitate better decision making on antimicrobial use [65].

To improve the quality of data collected, MAF is engaging with importers such as agriculture shops to understand their reservations on submitting import applications and exploring legislative tools to improve compliance on import application submission. Future monitoring could focus on collecting data more proximal to the site of usage such as at end-user level to elucidate species and production type usage patterns [83,84].

## 5. Conclusions

This baseline study demonstrated very low levels of antimicrobial use in animals in Timor-Leste consistent with its subsistence agriculture system. Antimicrobial classes imported in the largest quantities were tetracyclines, penicillins, and macrolides. This is very similar to usage patterns in other countries globally and regionally. Import of CIAs for administration via the oral route was high in the poultry industry, and antimicrobial use for growth promotion could not be ruled out. Antimicrobial use in the poultry industry is expected to rise due to industry growth and the limited alternative disease control strategies. Education of farmers, development of antimicrobial treatment guidelines and improving access to veterinary services can help to ensure good antimicrobial stewardship in the animal health sector. Through this study, in-country capacity to monitor antimicrobial imports according to OIE reporting requirements was developed.

## Figures and Tables

**Figure 1 antibiotics-10-00426-f001:**
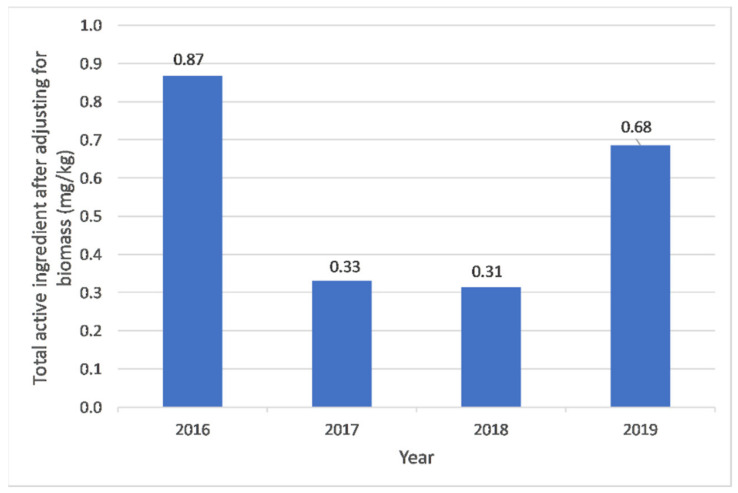
Antimicrobial import weight (mg) adjusted by animal biomass (kg) into Timor-Leste between 2016 and 2019.

**Figure 2 antibiotics-10-00426-f002:**
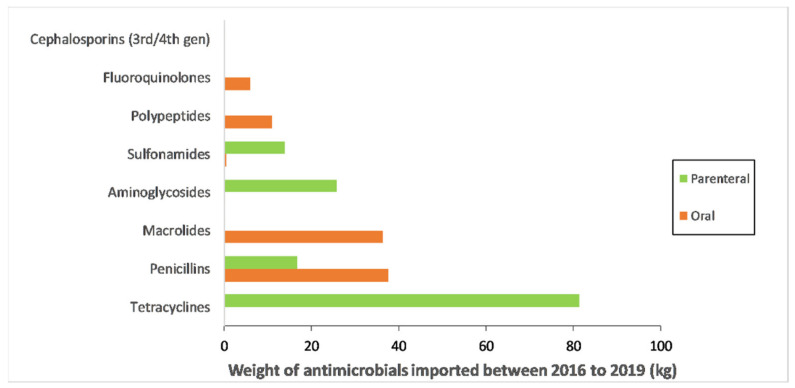
Total weight of veterinary antimicrobials imported into Timor-Leste between 2016 and 2019, by route of administration and antimicrobial class. Antimicrobials for administration via the topical route represented less than 0.3 kg (0.1%) of total imports and were therefore not included in the diagram.

**Figure 3 antibiotics-10-00426-f003:**
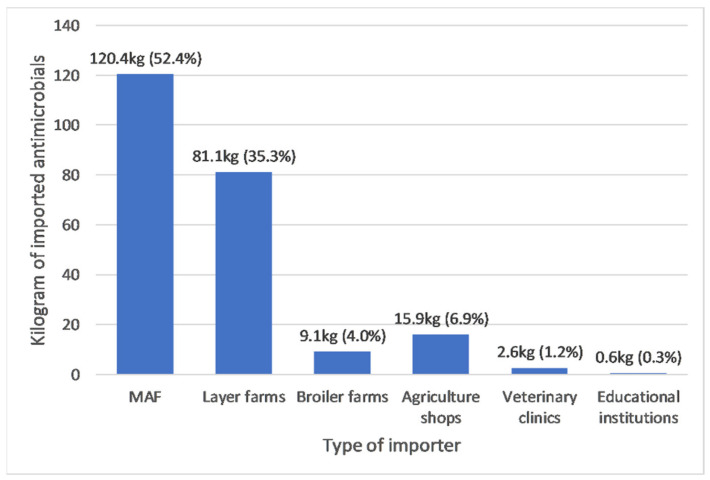
Total active ingredients imported between 2016 and 2019, by type of importer.

**Figure 4 antibiotics-10-00426-f004:**
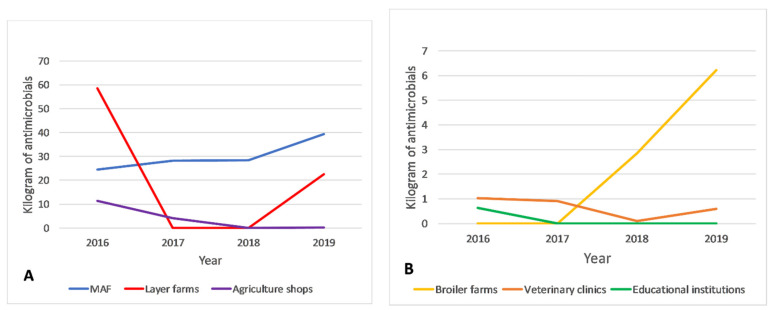
(**A**,**B**): Trend of antimicrobials imported by different importers between 2016 and 2019. Bigger importers are represented in (**B**) and smaller importers in (**B**), thus *y*-axes differ between diagrams.

**Figure 5 antibiotics-10-00426-f005:**
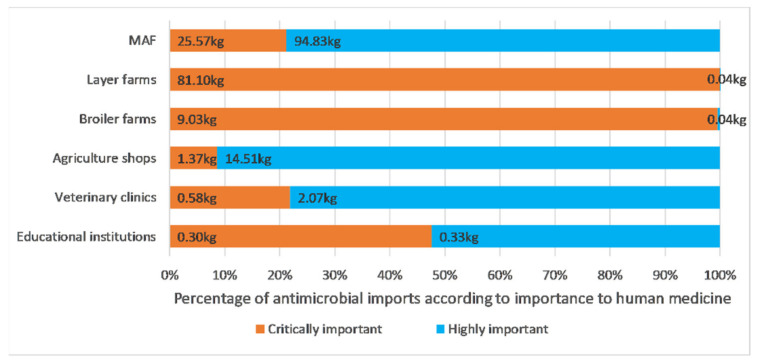
Profile of antimicrobial imports of different importer types by WHO classification of importance to human medicine. Important antimicrobials for human medicine represented less than 0.02 kg (0.01%) of total imports and were therefore not included in the diagram.

**Figure 6 antibiotics-10-00426-f006:**
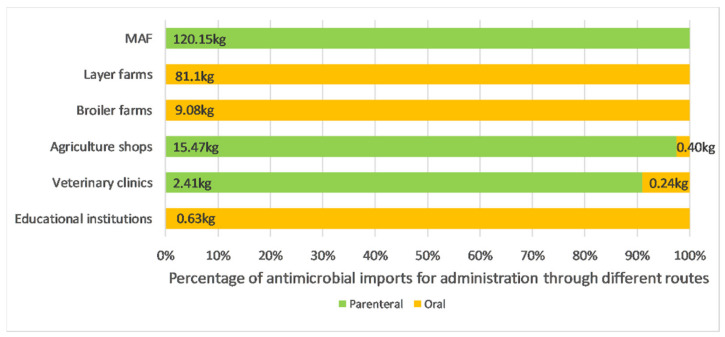
Profile of antimicrobial imports of different importer types, by recommended route of administration. Antimicrobials for administration via the topical route represented less than 0.3 kg (0.1%) of total imports and were therefore not included in the diagram.

**Table 1 antibiotics-10-00426-t001:** Weight of veterinary antimicrobials imported into Timor-Leste between 2016 and 2019, by year and overall for individual antimicrobial and antimicrobial class.

Antimicrobial Class	Antimicrobial (WHO Classification ^1^)	Kilogram of Active Ingredient (%)				r_s_ (*p* Value) ^2^	Kilogram for all Years (%)	Kilogram for all Years for Each Class (%)
		2016	2017	2018	2019			
Aminoglycosides	Neomycin (CIA)	0	2.5×10^−4^ (<0.01)	0.01 (0.05)	0.01 (0.02)	0.95 (0.05)	0.03 (0.01)	25.85 (11.25)
	Dihydrostreptomycin (CIA)	4.64 (4.83)	6.20 (18.56)	5.80 (18.5)	9.18 (13.32)	0.80 (0.20)	25.82 (11.24)	
Cephalosporins (3rd/4th gen)	Cefotaxime (CIA)	0	0.01 (0.03)	0	0	−0.26 (0.74)	0.01 (<0.00)	0.01 (<0.00)
Fluroquinolones	Norfloxacin (CIA)	0.20 (0.21)	0.20 (0.60)	0	0	−0.89 (0.11)	0.40 (0.17)	6.01 (2.62)
	Enrofloxacin (CIA)	3.61 (3.76)	0	0	2.00 (2.90)	−0.32 (0.68)	5.61 (2.44)	
Macrolides	Tylosin (CIA)	25.10 (26.12)	0.10 (0.30)	0	0	−0.95 (0.05)	25.20 (10.97)	36.45 (15.86)
	Tilmicosin (CIA)	0	0	0	11.25 (16.33)	0.77 (0.23)	11.25 (4.9)	
Penicillins	Ampicillin (CIA)	0	0	3.27 (10.41)	0.63 (0.91)	0.74 (0.26)	3.89 (1.69)	54.39 (23.67)
	Benzylpenicillin (HIA)	2.78 (2.90)	3.72 (11.13)	3.48 (11.10)	5.70 (8.28)	0.80 (0.20)	15.69 (6.83)	
	Amoxicillin (CIA)	20.31 (21.13)	0	0	14.51 (21.05)	−0.32 (0.68)	34.82 (15.15)	
Polypeptides	Bacitracin (IA)	0	0	0.02 (0.06)	0	0.26 (0.74)	0.02 (0.01)	10.93 (4.76)
	Colistin (CIA)	10.04 (10.45)	0	0.31 (0.98)	0.57 (0.82)	−0.20 (0.80)	10.91 (4.75)	
	Polymyxin B (CIA)	0	9.5 × 10^−5^ (<0.01)	0	0	−0.26 (0.74)	9.5 × 10^−5^ (<0.01)	
Sulfonamides	Sulfamonomethoxine (HIA)	0.40 (0.42)	0.20 (0.60)	0	0	−0.95 (0.05)	0.60 (0.26)	14.46 (6.29)
	Sulfaquinoxaline (HIA)	0.33 (0.34)	0	0	0	−0.77 (0.23)	0.33 (0.14)	
	Sulfadoxine (HIA)	0	0	0	0.04 (0.06)	0.77 (0.23)	0.04 (0.02)	
	Sulfamerazine (HIA)	0	0.08 (0.23)	0.02 (0.06)	0.78 (1.14)	0.80 (0.20)	0.88 (0.38)	
	Sulfadiazine (HIA)	2.09 (2.17)	1.48 (4.42)	1.59 (5.06)	1.57 (2.27)	−0.40 (0.60)	6.72 (2.92)	
	Sulfadimidine (HIA)	1.67 (1.74)	0.58 (1.73)	0.80 (2.56)	2.35 (3.41)	0.40 (0.60)	5.40 (2.35)	
	Trimethoprim (HIA)	0.20 (0.21)	0.28 (0.84)	0	0.01 (0.01)	−0.06 (0.40)	0.49 (0.21)	
Tetracyclines	Oxytetracycline (HIA)	24.73 (25.73)	20.56 (61.55)	16.07 (51.23)	20.32 (29.48)	−0.80 (0.20)	81.67 (35.54)	81.67 (35.54)
Overall		96.1 (100)	33.41 (100)	31.36 (100)	68.91 (100)	−0.40 (0.60)	229.77 (100)	229.77 (100)

^1^ CIA refers to critically important antimicrobials; HIA refers to highly important antimicrobials; IA refers to important antimicrobials.^2^ Spearman rank-order correlation coefficient (r_s_) and *p*-value assessing the strength and direction of possible monotonic trends in the quantities of antimicrobials imported over time.

## Data Availability

The data presented in this study are available in the Appendix A.

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
