# Peer review of "Antimicrobial Use in Animals in Timor-Leste Based on Veterinary Antimicrobial Imports between 2016 and 2019"

_antibiotics, 2021, doi:10.3390/antibiotics10040426_

Round 1

Reviewer 1 Report

Accurate information on the use of antimicrobial agents at the country and global level is crucial for informing strategies to combat antimicrobial resistance. Previously, the calculation of antimicrobial agents intended for use in animals in Timor-Leste was conducted on an ad hoc basis without consistent methods. Hence, this article should be published.

Minor comments:

  • quality of figures should be improved, the letters and numbers are too small,
  • fig. 1-3 - error bars should be added,
  • of course the data are well presented, however the Author could try to use chemometry or statistics, ex. network for data visualization.

Reviewer 2 Report

The study was aimed to monitor the veterinary antimicrobial use in Timor-Leste between 2016 and 2019, describing the type used, differents species in which are applied, and the administration routes.

In my opinion, as described by the authors, the period of the study is very short, so data are limited to may formulate salient conclusions.

The paragraph regarding the antimicrobial class and importance for human medicine should be necessarily amplified. 

Round 2

Reviewer 2 Report

Changes required have been performed. Thank you.